# Who Benefits Most from Positive Psychological Interventions? Predictors and Moderators of Well-Being Outcomes in Severe Mental Health Conditions

**DOI:** 10.3390/healthcare13161988

**Published:** 2025-08-13

**Authors:** Regina Espinosa, Almudena Trucharte, Alba Contreras, Vanesa Peinado, Carmen Valiente

**Affiliations:** 1Department of Psychology, Faculty of Health Sciences-HM Hospitals, Camilo José Cela University, 28692 Madrid, Spain; almutruc@ucm.es; 2HM Hospitals Health Research Institute, 28015 Madrid, Spain; 3Department of Psychobiology and Methodology of Behavioural Sciences, University of Málaga, 29071 Málaga, Spain; albacont@ucm.es; 4Department of Clinical Psychology, School of Psychology, Complutense University, 28223 Madrid, Spain; juliavpe@ucm.es (V.P.); mcvalien@ucm.es (C.V.)

**Keywords:** positive psychology interventions, severe psychiatric conditions, treatment moderators

## Abstract

**Background/Objectives**: Positive psychology interventions (PPIs) may enhance well-being in individuals with severe psychiatric conditions (SPCs), yet little is known about individual differences in treatment response. **Methods**: We conducted a secondary analysis of a single-blind, parallel-group randomized controlled trial. A total of 119 adults receiving outpatient mental health care were randomized to an 11-week multicomponent PPI plus treatment as usual (PPI + TAU) or TAU alone. A priori demographic and baseline clinical variables (e.g., age, gender, education, diagnosis, symptom severity) were tested as predictors and moderators of six well-being outcomes. Moderation analyses were conducted using the PROCESS macro (version 4.1) for SPSS version 29.0, with simple slopes explored for significant interactions. Analyses followed an intention-to-treat approach. **Results**: Individuals who were unemployed, had a diagnosis within the psychosis spectrum, or exhibited high interpersonal sensitivity showed improvements in well-being irrespective of the treatment modality received. Older patients, those attending more weekly therapy sessions, and individuals with less somatization, hostility, or life satisfaction levels responded particularly well to the specialized PPI + TAU treatment. While several interactions were significant at *p* < 0.01, none remained significant after Bonferroni–Holm correction. Nevertheless, the patterns were consistent and theoretically grounded. **Conclusions**: Individual characteristics may influence the effectiveness of PPIs in SPC populations. Identifying predictors and moderators can inform more personalized interventions. The findings warrant replication. Trial registration: ClinicalTrials.gov, NCT01436331.

## 1. Introduction

Well-being, as a distinct concept of physical and mental health conditions [1], encompasses both hedonic dimensions (subjective happiness, positive affect, and life satisfaction) and eudaimonic aspects (psychological functioning including purpose, personal growth, autonomy, and meaningful relationships). It is gaining significant relevance due to its profound relationship with mental health recovery processes [2,3]. Acting as a protective factor against physical and psychological disorders [4], its impact on improving mental health symptoms in severe psychiatric conditions (SPCs), including schizophrenia spectrum disorders, bipolar disorder, severe recurrent depression, and personality disorders that substantially limit social, occupational, or interpersonal functioning, has positioned well-being as a key target in psychological interventions [5].

Multiple therapeutic modalities incorporate strength-based principles. That is, methods based on simultaneously cultivating client assets while addressing psychological difficulties and symptomatology [6]. These therapeutic modalities, including positive psychology interventions (PPIs, e.g., gratitude-based interventions or strengths spotting exercises) [7,8], resilience-enhanced cognitive-behavioral interventions (e.g., problem-solving skills training, -based stress reduction (MBSR) or acceptance and commitment therapy (ACT) techniques) [9,10,11]. In this context, PPIs based on standardized exercises such as gratitude exercises, identification and application of character strengths, enjoyment techniques, loving-kindness meditation, and “best possible self” writing exercises [12,13,14], have demonstrated their effectiveness in increasing well-being among individuals with psychosis and SPCs through multiple studies [15,16,17,18]. Metanalytic evidence further confirms that people with SPCs benefit from PPIs in terms of enhanced mental health [19]. Also, mindfulness-based interventions (MBIs) have garnered substantial empirical support, with multiple recent meta-analyses indicating significant improvements in psychosocial functioning, insight, negative symptoms, and overall symptomatology when combined with standard care [20,21,22]. These approaches offer complementary therapeutic mechanisms: PPIs target enhancement of positive emotions, character strengths, and meaning-making processes [13], while MBIs facilitate metacognitive awareness and non-reactive observation of internal experiences, reducing attachment to distorted thought processes [23]. ACT emphasizes psychological flexibility through acceptance of difficult internal experiences while promoting values-based behavioral engagement [24]. Emerging evidence suggests synergistic effects between these modalities, particularly how mindfulness practices may enhance receptivity to PPIs by increasing present-moment awareness and reducing reactivity to negative mental events, thereby creating optimal conditions for cultivating positive emotions and engaging in meaningful activities [25,26]. Also, integrating these interventions within mental health services has shown the potential to foster recovery by developing resources that promote personal well-being and manage overall symptomatology [27,28]. Consequently, the literature substantiates that implementing PPIs for individuals with psychosis is both feasible and effective in improving well-being and its dimensions [29].

In general, while evidence-based practice significantly advanced research on psychological treatments and their implementation in clinical practice [30], the variability in the outcomes among individuals receiving psychological interventions has led to an increasing focus on personalized treatments [31,32]. Psychological interventions rarely produce uniform effects across individuals. Differences in participants’ characteristics, the context of intervention delivery, and the fidelity of implementation can result in varying levels of effectiveness. Examining these individual differences in the treatment responses provides valuable insights into the underlying mechanisms of interventions, enabling improvements in treatment designs to enhance their efficacy for a broader range of participants [33]. In this sense, certain interventions would benefit some individuals more than others due to specific characteristics or pre-existing variables. While PPIs are generally effective interventions for enhancing well-being, the specific factors that might help explain individual differences in this efficacy are underexplored [7,33]. This raises an important question: is there any characteristic that might explain why some individuals respond better to well-being interventions than others?

The existing literature has highlighted prognostic (also known as. predictor variables that might anticipate treatment’ responses) and prescriptive factors (i.e., moderators that might predict differential responses to interventions) [34,35]. For instance, research on well-being suggests that different sociodemographic and clinical factors may be associated with the presence of well-being as predictors, while other factors may exert differential effects on the response to treatment as moderators. Research on gender differences in well-being is heterogeneous across dimensions of well-being. In general, studies indicate that women tend to have lower overall levels of well-being than men. However, they may show similar a responsiveness to well-being interventions as men, as is the case with interventions for depression [36]. This led us to investigate whether sex differences influenced individual responses to our well-being intervention. Similarly, education and income have been associated with general well-being [37,38]. Thus, while these variables could affect the effectiveness of PPIs, previous studies have not yielded conclusive results regarding the moderating effects on response to well-being intervention of other variables such as baseline levels of premorbid functioning [39], medication adherence [32], diagnosis of non-affective psychosis [33] higher baseline symptom severity [7,39] and positive affect [7,40], so we will explore both possibilities in our analyses.

Therefore, this study represents a secondary analysis of data from Valiente et al. [18,41]. The original study was designed to explore the efficacy and effectiveness of a multicomponent positive psychology intervention to improve well-being for individuals with SPCs in comparison with treatment as usual (TAU). The current analysis was designed to address distinct research questions aimed at identifying potential predictors and moderators of a positive response to treatment. We sought to identify demographic and baseline clinical characteristics that might act as predictors and/or moderators of treatment outcomes (e.g., well-being) of an 11-week PPI + TAU intervention for individuals with SPC.

Based on the existing literature, we hypothesized that, regardless of treatment (PPI + TAU intervention or treatment as usual, TAU), post-intervention well-being levels would be higher in individuals with a later age at onset, with higher levels of education, and with a diagnosis of non-affective psychosis and a lower baseline symptom severity. Given the inconclusive information about gender and moderators, no specific a priori hypotheses were formulated; thus, moderator analyses will be exploratory and use the same set of variables analyzed as predictors. We used the same participant cohort but with different primary outcomes and analytical approaches.

## 2. Materials and Methods

### 2.1. Participants

We used participant data from a larger project (for details, see [18,41]) involving a randomized controlled trial to test the efficacy of a multicomponent PPI + TAU vs. a wait-list condition (WL) with TAU to improve well-being in people with SPC (Trial registration: ClinicalTrials.gov, NCT01436331). One hundred and forty-two participants were initially assessed for eligibility and provided consent to participate in the study. After excluding one participant who failed to complete the intake assessment, 141 participants were randomly allocated to either the PPI + TAU group (n = 71) or the WL group (n = 70). However, 22 participants did not complete their post-assessment evaluation.

The final sample consisted of 119 individuals with severe psychiatric conditions (SPCs), referred to the study by their key therapist if they met the following inclusion criteria: (a) 18–65 years old and (b) demonstrated minimal motivation and commitment to participate in group therapy. Participants were excluded, if they had (a) limited cognitive resources or a severe formal thought disorder and (b) a concurrent condition (i.e., a current diagnosis of substance dependence or a severe personality disorder with psychosis or affective disorder) that could interfere with a psychotherapy group (see [18,41] for further details about the randomized control trial procedure). Demographic and clinical variables and mean scores for the psychological variables used as potential predictors or moderators of treatment outcomes are presented in Table 1 and Table 2.

### 2.2. Intervention Conditions

The experimental treatment consisted of a multi-component group intervention based on PPI and ACT techniques adjunctive to routine care (treatment as usual, TAU: individual psychopharmacological and therapeutic interventions and individualized use of psychosocial resources). The group intervention was structured as a three-module progressive program addressing (1) emotional regulation and awareness, (2) self-acceptance and self-compassion, and (3) values clarification and life purpose identification [18]. Drawing from Fredrickson’s [42] broaden-and-build theory, the intervention design followed a developmental trajectory from hedonic well-being components toward eudaimonic functioning enhancement. While the protocol acknowledged the specific challenges and recovery contexts inherent to SPCs, the primary therapeutic targets were positive psychological processes rather than symptom reduction, specifically focusing on cultivating positive emotional experiences, strengthening adaptive self-relationships and interpersonal connections, activating personal strengths and resources, and clarifying meaningful values and life purposes.

The intervention protocol integrated evidence-based elements from multiple therapeutic modalities. Building upon established PPI research with schizophrenia populations [14], core components included standardized exercises of positive emotion cultivation, gratitude practice, forgiveness work, and systematic strengths identification and application. Self-compassion and positive self-care strategies derived from Gilbert’s [43]. Compassion-Focused Therapy was incorporated to address self-critical patterns common in this population. Additionally, key ACT principles adapted for psychosis [11] were integrated, particularly psychological acceptance processes and values-based goal formation, enabling participants to develop behavioral patterns congruent with their identified personal values and life directions.

PPI + TAU were delivered in 11 weekly, 90 min sessions of group intervention. For a more detailed description, please refer to the protocol and the manual [18,29]. Each group was facilitated by two therapists who possessed clinical expertise in conducting psychotherapy groups for individuals with SPCs, averaging 11.9 years of experience. The majority of therapists were psychologists (n = 23), while seven were social workers. To ensure protocol adherence and address session challenges, all group therapists participated in regular one-hour supervision sessions held bimonthly, which provided ongoing support and additional guidance for implementing the protocol exercises. The control group, which was on a waiting list, received TAU only. TAU primarily consists of personalized outpatient care, including psychopharmacological management and individual psychological therapy. The focus is on symptom stabilization, relapse prevention, and functional recovery. This integrative approach typically draws on cognitive-behavioral therapy, psychoeducation, and recovery-oriented models. Sessions are typically scheduled on a biweekly or monthly basis, depending on clinical needs and availability.

### 2.3. Instruments and Variables

To measure the effect of the intervention on well-being levels, we evaluated the changes found in all dimensions of eudaimonic well-being by calculating the difference between post-intervention and baseline well-being levels. That is, higher scores on this index indicated that well-being levels had increased after the intervention, and negative scores indicated that well-being levels had worsened relative to their baseline. Two independent evaluators administered the Scales of Psychological Well-Being (SPWB; 36 items) [44], which measure six dimensions: self-acceptance, positive relations with others, autonomy, environmental mastery, purpose in life, and personal growth. The internal consistency of these subscales in the current study ranged from acceptable to good (α =  0.84, 0.79, 0.69, 0.83, 0.75, and 0.77, respectively). The SPWB was administered twice: once at baseline (pre-assessment) and again 11 weeks later (post-assessment).

As potential predictors or moderators, we included both sociodemographic and clinical baseline characteristics. These included gender, age, civil status, educational level, employment status, age at first symptom onset, number of individual therapy sessions, and alcohol consumption.

We also assessed baseline levels of psychopathological symptoms using the Symptom Checklist-90-Revised (SCL-90-R) [45], which includes the following subscales: Somatization, Obsessive–Compulsive, Interpersonal Sensitivity, Depression, Anxiety, Anger-Hostility, Phobic Anxiety, Paranoid Ideation, and Psychoticism. In our study, these subscales showed good internal consistency, with Cronbach’s alpha values of 0.87, 0.88, 0.76, 0.89, 0.85, 0.90, 0.84, 0.81, and 0.90, respectively. The SCL-90-R was administered twice: once at baseline (pre-assessment) and again 11 weeks later (post-assessment).

Additionally, baseline levels of satisfaction with life were measured using the Satisfaction With Life Scale (SWLS) [46], which also demonstrated good internal consistency (Cronbach’s α = 0.82). The SWLS was administered twice: once at baseline (pre-assessment) and again 11 weeks later (post-assessment).

We decided not to include any well-being dimension-related moderator variables in the analyses, as this would complicate the interpretation of results because of the high correlation with the outcome variables and regression to the mean.

### 2.4. Statistical Analyses

To identify predictors and moderators of treatment outcomes, we selected a subset of baseline sociodemographic and clinical variables a priori based on clinical and empirical rationale, including data from the literature. We performed a simple moderation analysis using the PROCESS macro (version 4.1) for SPSS version 29.0 [47] to examine whether sociodemographic and baseline clinical variables moderated the relationship between the treatment group (PPI + TAU vs. TAU; the independent variable) and changes in well-being dimensions (the dependent variables) post-treatment. Specifically, we tested whether the strength of this relationship varied as a function of a third variable acting as a moderator. In the case of statistically significant interactions, we conducted tests of the simple slopes for the well-being dimensions conditioned at each of the levels of moderators [48]. Following this procedure, we selected a value, M, calculated the conditional effect of X on Y (θX→Y) at that value, and conducted an inferential test (see [48]). When the main effect of a variable was significant, but the interaction was not, the variable was considered a non-specific predictor of the outcome. Conversely, when the interaction was significant (regardless of the significance of the main effects), the variable was considered a moderator. Given the large number of moderation tests (n = 120) and the a priori selection of moderators and outcome variables based on theoretical and empirical grounds, no formal correction for multiple comparisons was applied. Instead, the results are interpreted in light of the confirmatory nature of the hypotheses tested [49,50]. *p*-values are reported for transparency, but caution is advised when interpreting marginal effects.

## 3. Results

The results highlighted several demographic and clinical variables as potential predictors of treatment outcome, independent of treatment allocation. Of the 120 moderation analyses conducted, 4 yielded statistically significant interaction effects (*p* < 0.05). These included age, number of weekly therapy sessions, and somatization, hostility, or life satisfaction baseline levels. Although these *p*-values did not survive correction for multiple comparisons (e.g., Bonferroni-adjusted *α* = 0.00042), they may still be meaningful given the a priori hypotheses and the consistency of effects across related outcomes. Simple slopes analyses were conducted for significant interactions, following the procedure of Aiken and West [48].

### 3.1. General Predictors of Treatment Response

Among the sociodemographic characteristics, being unemployed was associated with greater increases in the autonomy well-being dimension at post-treatment. In addition, a psychosis spectrum diagnosis appeared to predict greater improvements in the post-treatment well-being dimensions of positive relationships with others and purpose in life. Finally, higher baseline levels of interpersonal sensitivity were associated with greater increases in the levels of self-acceptance after treatment (see Appendix A in the Appendix A).

### 3.2. Moderators of Differential Treatment Response

We identified several moderators of treatment response in certain well-being dimensions, depending on the treatment condition (see Appendix A in Appendix A). Firstly, the results indicated a moderation effect of age on personal growth (*b* = 0.32, *p* = 0.002). In particular, the PPI + TAU condition appeared to be more beneficial than TAU alone with increasing age. Further analyses revealed significant differences between age groups, with specific cut-off points at 33, 43 and 52 years (see Figure 1). The experimental treatment (PPI + TAU) showed greater efficacy than TAU for individuals aged 52 and older. However, for the younger age group (33 years), the TAU condition showed superior results compared to the experimental treatment.

Secondly, given that all participants received individual psychological sessions as part of the TAU component, the analysis revealed differences in treatment responses based on the number of individual psychological sessions per week. Specifically, the PPI + TAU condition was more effective in increasing self-acceptance levels compared to TAU alone when participants attended more weekly therapy sessions (*b* = 4.49, *p* = 0.007). The analysis identified specific cut-offs for the number of sessions per week (0.25, 0.50 and 1) at which the experimental treatment began to show a significantly greater effect than TAU (see Figure 2). Notably, attending one session per week significantly increased self-acceptance after treatment.

Thirdly, the findings revealed baseline levels of two SCL-90 dimensions (somatization and hostility) as potential moderators for three well-being dimensions (autonomy, positive relationships with others and personal growth). Specifically, the experimental treatment (PPI + TAU) was more effective in increasing autonomy levels compared to TAU when baseline levels of somatization were lower (*b* = −2.89, *p* = 0.01). The analysis identified specific cut-offs for somatization levels (0.25, 0.83 and 1.91), showing that lower levels of somatization (0.25 and 0.83) had a significant positive effect on autonomy in the PPI + TAU condition (see Figure 3). Furthermore, the results indicated a moderate effect of baseline hostility levels on positive relationships with others (*b* = −2.71, *p* = 0.03). The experimental treatment (PPI + TAU) was more beneficial than TAU when hostility levels were lower. Further analyses identified specific cut-off points (0, 0.33 and 1.5) for hostility levels, showing that PPI + TAU tended to be more effective (though not significantly so) at the lowest hostility level (0), whilst TAU tended to be more effective (though not significantly so) at the highest hostility level (1.5). Hostility also moderated personal growth (*b* = −3.30, *p* = 0.007), with TAU being more beneficial at the highest hostility level (1.5) (see Figure 4). Finally, the results indicated a moderation effect of baseline SWLS levels on environmental mastery (*b* = −0.43, *p* = 0.004). In particular, the experimental treatment (PPI + TAU) was more beneficial than TAU when baseline SWLS levels were lower. Subsequent analyses identified significant cut-off points for SWLS levels (11.2, 17 and 25), with the experimental treatment being significantly superior to TAU at lower SWL levels (11.2 and 17) (see Figure 5).

In summary, the results showed that older individuals, those who attended a greater number of individual psychotherapy sessions per week and had lower baseline levels of somatization, hostility and SWLS, experienced greater increases in most dimensions of well-being after 11 weeks when receiving the PPI + TAU intervention. However, all effect sizes calculated for the predictors and moderators ranged from small to medium according to Cohen’s criteria (see Appendix A in Appendix A).

## 4. Discussion

This study examined theoretically grounded predictors and moderators of treatment outcomes in individuals receiving care for SPC.

Our findings provide valuable insights into the variability of PPI + TAU effectiveness across distinct patient subgroups. Identifying general predictors of treatment outcome offers essential prognostic information, supporting clinicians in identifying individuals who are more likely to benefit from interventions overall, regardless of treatment type. Notably, patients who were unemployed, diagnosed with a psychosis spectrum disorder, or presented higher baseline levels of interpersonal sensitivity reported better well-being outcomes at 11 weeks, independent of treatment condition. Interestingly, these findings contrast with previous literature, which has suggested that individuals with less chronic conditions and later age of first contact with mental health services tend to respond more favorably to both pharmacological and psychological interventions [51,52]. It is possible that individuals with more complex psychological profiles, who are frequently unemployed, may demonstrate greater adherence to and engagement with psychosocial resources, thus benefiting more substantially from interventions focused on psychological well-being. In contrast, no predictive effects were found for gender or educational level. This is consistent with meta-analytic evidence suggesting that sociodemographic factors generally do not show strong associations with treatment outcomes [53,54].

On the other hand, treatment moderators offer prescriptive value, informing which individuals may benefit most from specific interventions [55]. While predictors guide general clinical expectations [34], understanding moderators enhances personalized treatment planning and cost-effectiveness. Although several interaction terms reached conventional significance thresholds (e.g., *p* < 0.01), these did not remain statistically significant after applying strict corrections for multiple comparisons (e.g., Bonferroni-adjusted *α* = 0.00042). However, given the a priori selection of both moderators and outcome variables based on theoretical and empirical justification, it is defensible not to apply overly conservative corrections [49,50]. Thus, the results should be interpreted as preliminary yet potentially meaningful patterns that warrant replication.

In this context, we identified several significant moderators of treatment outcomes. Age moderated the effects of PPI + TAU on personal growth, with older participants showing greater improvements. This may be explained by age-related advantages in emotional regulation, introspection, and receptivity to personal development [56]. However, the relationship was non-linear, highlighting the nuanced relationship between age and intervention efficacy. Another relevant moderator was the frequency of individual psychological sessions as part of TAU. Specifically, attending at least one session per week appeared to enhance the effectiveness of the group PPI + TAU, particularly in self-acceptance, suggesting that a minimum therapeutic dose may be necessary to support the integration of the intervention’s content.

Baseline levels of somatization and hostility also acted as moderators. Participants with lower somatization levels in the PPI + TAU group demonstrated greater improvements in autonomy, while hostility showed a complex interaction with outcomes in positive relationships and personal growth. These findings are in line with previous studies that suggest individuals with lower baseline psychopathology are more responsive to psychosocial interventions [52,57,58]. Somatization may moderate PPI + TAU efficacy because individuals with high somatization often have difficulty engaging in cognitive-emotional tasks, limiting their ability to benefit from interventions like PPI+ TAU that require reflection and emotional awareness. Lower somatization levels may allow greater engagement, particularly in outcomes like autonomy. Lower hostility levels may reflect greater emotional regulation and interpersonal openness, facilitating engagement with the intervention and promoting positive change. Individuals with lower hostility may be more receptive, enhancing gains in positive relationships and personal growth. Furthermore, life satisfaction at baseline moderated changes in environmental mastery, with the PPI + TAU group showing greater gains among individuals with initially low satisfaction, possibly due to increased motivation for improvement. These findings underscore the relevance of individual psychological profiles in predicting intervention outcomes. While positive psychology interventions may support the personal recovery of individuals with psychosis, demonstrating their sustained impact on well-being and symptomatology in this population remains particularly challenging.

Despite the promising results, several limitations must be acknowledged. First, this analysis shares participants and data collection methods with previous publications [18,41], potentially limiting the independence of findings. Nonetheless, the current study addressed distinct research questions using different analytical approaches, offering novel insights into individual variability in treatment responses. Second, the specific inclusion criteria used for this study and clinical setting in which it was conducted, may limit the generalizability of the findings, as could the number of male (almost 60%) and single participants. Third, although therapist training and fidelity to the intervention protocol were carefully monitored, we did not assess therapist skills or therapeutic alliance, which are known to influence psychotherapy outcomes [59]. Also, the relatively short follow-up period (11 weeks) precludes conclusions about the durability of the observed effects, and unmeasured confounding variables such as life stressors occurring during the treatment period, medication adherence patterns, interventions received outside the study, may have influenced results. Finally, while our randomization ensured a balanced diagnostic composition between groups, unmeasured variables such as specific medication regimens and dosages may have influenced treatment responses and warrant investigation in future studies.

## 5. Conclusions

In conclusion, this study emphasizes the differential effectiveness of positive psychotherapy integrated with treatment as usual in patients with severe psychiatric conditions. By identifying both predictors and moderators of treatment response, our findings contribute to advancing personalized psychological care. Tailoring interventions in terms of content, intensity, and timing may further optimize their impact. Understanding individual differences in response not only informs clinical decision-making but also sheds light on the underlying mechanisms of therapeutic change, paving the way toward more targeted and efficient mental health interventions.

## Figures and Tables

**Figure 1 healthcare-13-01988-f001:**
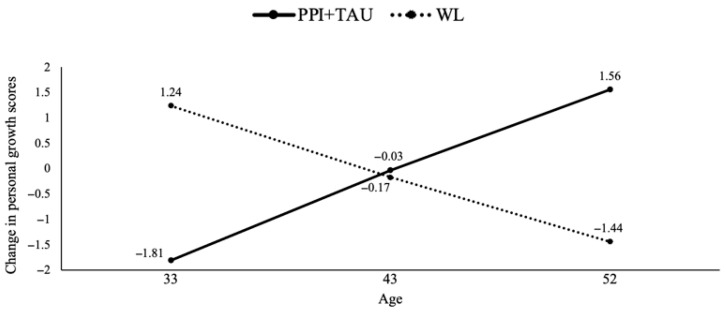
Treatment group differences in personal growth changes moderated by age. Note. PPI = positive psychological intervention; TAU = treatment as usual; WL = waiting list.

**Figure 2 healthcare-13-01988-f002:**
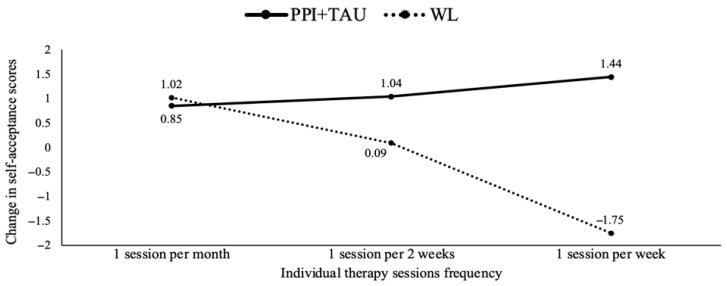
Treatment group differences in Self-acceptance changes moderated by individual therapy sessions frequency. Note. PPI = positive psychological intervention; TAU = treatment as usual; WL = waiting list.

**Figure 3 healthcare-13-01988-f003:**
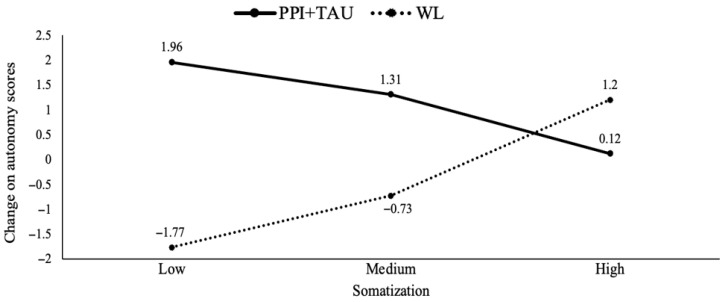
Treatment group differences in autonomy changes moderated by different baseline somatization levels. Note. PPI: positive psychological intervention; TAU: treatment as usual; WL: waiting list.

**Figure 4 healthcare-13-01988-f004:**
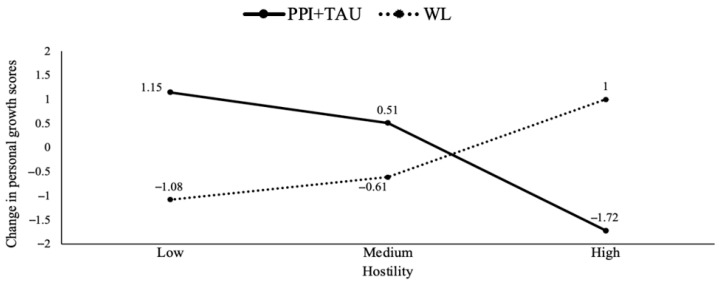
Treatment group differences in personal growth changes moderated by different baseline hostility levels. Note. PPI: positive psychological intervention; TAU: treatment as usual; WL: waiting list.

**Figure 5 healthcare-13-01988-f005:**
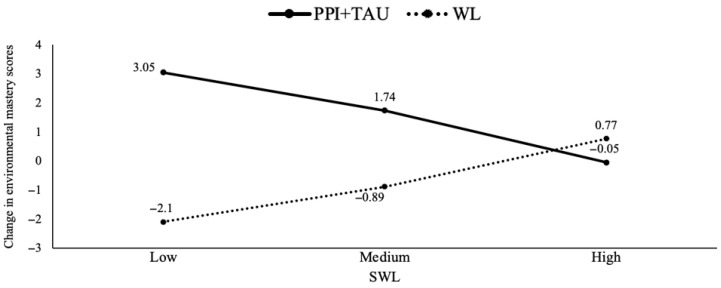
Treatment group differences in environmental mastery changes moderated by different baseline life satisfaction levels. Note. PPI: positive psychological intervention; SWLS: satisfaction with life; TAU: treatment as usual; WL: waiting list.

**Table 1 healthcare-13-01988-t001:** Participants’ demographic and clinical characteristics (N = 119).

Age in years, mean (SD)	42.7 (9.5)
Sex: Men, n (%)	71 (59.7)
Single status, n (%)	105 (88.2)
Education, n (%)	
Elementary school	28 (23.9)
Secondary School	54 (46.2)
College Education	34 (29.1)
Employed, n (%)	
Unemployed	105 (89.7)
Part-time employment	11 (9.4)
Full-time employment	1 (0.9)
Diagnosis, n (%)	
Schizophrenia	82 (71.3)
Affective disorders	12 (10.4)
Anxiety disorders	7 (6.1)
Personality disorders	9 (7.8)
Others	5 (4.3)
Medication, n (%)	
Benzodiazepines	72 (69.9)
Hypnotics (No benzo)	4 (4.2)
Antipsychotics	99 (89.2)
Anti-depressants	43 (45.3)
Mood Stabilizers	23 (23.7)
First psychiatric symptoms, n (%)	
Childhood	1 (0.8)
Adolescence	24 (20.2)
Adulthood	66 (55.5)
Therapy frequency, n (%)	
No therapy	6 (5.2)
Hour per week	48 (41.4)
1 h each 2 weeks	35 (30.2)
1 h per month	23 (19.8)
Less	4 (3.4)

Note. SD = standard deviation.

**Table 2 healthcare-13-01988-t002:** Participants’ mean scores on the psychological variables (N = 119).

SPWB, mean (SD)	
Autonomy	19.7 (4.27)
Environmental mastery	31.0 (7.59)
Personal growth	35.6 (7.31)
Positive relationships	34.3 (7.90)
Purpose in life	34.4 (7.30)
Self-acceptance	30.9 (8.27)
SWLS, mean (SD)	17.9 (7.04)
SCL-90, mean (SD)	
Anxiety	1.23 (0.89)
Depression	1.50 (0.89)
Interpersonal sensibility	1.43 (0.88)
Paranoid Ideation	1.23 (0.90)
Somatization	1.02 (0.81)
Hostility	0.70 (0.80)
Compulsion/obsession	1.66 (0.89)
Phobic Anxiety	1.03 (0.91)
Psychoticism	1.08 (0.83)

Note. SD = standard deviation; SPWB: scales of psychological well-being; SWLS: satisfaction with life scale. SCL-90-R: Symptom Checklist-90-R.

## Data Availability

The data presented in this study are available on request from the corresponding author.

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
