# Peer review of "Who Benefits Most from Positive Psychological Interventions? Predictors and Moderators of Well-Being Outcomes in Severe Mental Health Conditions"

_healthcare, 2025, doi:10.3390/healthcare13161988_

Round 1
Reviewer 1 Report
Comments and Suggestions for Authors
- Introduction
- Line 1 revise "regarding to identify" to "aimed at identifying"
- Line 47 should be "positive psychology interventions (PPIs)"
- Line 52 comma after (7,9)
- Line 79 "predictors" should be "predictor"
- Line 99-101 Rewrite. For example "The current analysis was designed to address distinct research questions aimed at identifying potential predictors and moderators of a positive response to treatment."
- Line 103 change "a" to "an"
- I would include clearer definitions of well-being and what are considered severe psychiatric conditions. In addition, what types of interventions are considered PPIs, resilience-based, strength-based, and resource-activated?
- Materials and Methods
- Line 118 if you begin a sentence with a number, spell it out.
- How did you differentiate between severe personality disorders and personality disorders that were included?
- I would create different tables for the demographics and the instrument scores. It is confusing to be reading demographic information and then see scores that we don't yet understand or know what we are looking at.
- Line 149 what acceptance and commitment techniques? This is the first mention of ACT. It needs to be explained in the introduction. What PPI techniques were used? You refer to the manual, but it would be very helpful to include a brief explanation here.
- Line 160 remove comma after "which"
- Lines 167 and 169 repeat the same idea/phrasing. Revise this.
- Lines 174-178 rewrite this sentence into multiple sentences. This is a confusing sentence covering multiple ideas.
- I would include the internal consistency of the scales with a separate description for clarity. Then explain the process for administration. This section is confusing and hard to read/understand.
- Results
- Remove lines 214-216
- Line 240 How did you determine the age cut-offs?
- Line 284 change "moderatos" to "moderators"
- Lines 283-303 what is your rationale for expecting hostility and somatization to moderate PPI efficacy?
- Line 297 reword "being TAU more beneficial" to "with TAU being more beneficial"
- remove comma after identify
- Figures 3, 4, and 5 need more of a description of the data and what we are looking at.
- You refer to the Satisfaction With Life Scale as SWLS at times and SWL at others (lines 298-301). Change this to reflect consistent abbreviation.
- You only report p-values, not effect sizes. The effect sizes based on Cohen's d or beta coefficients can provide meaningful context even when the p-values are not significant.
- Discussion
- Lines 362-365 reword to “This study examined theoretically grounded predictors and moderators of treatment outcomes in individuals receiving care for SPC.”
- Lines 397-398 consider rephrasing to “highlighting the nuanced relationship between age and intervention efficacy."
- Line 400 you are inconsistent with using PPI+TAU vs. just PPI. You should be consistent with how you refer to the treatment condition.
- Integrate lines 414-417 into one of the other paragraphs.
- Lines 427-428 could benefit from some specific examples of possible confounding variables.
- Conclusions
- Lines 430-431 you spell out your PPI and TAU but use the abbreviation for SPC. I would spell out SPC here as well.
I would spell out abbreviations the first time they appear in each section. This helps remind the reader of their meaning without having to scroll back to the beginning. More time/space should be given to what the assessments are and what they test.
The reference list has some major APA formatting issues.
Overall, this is a strong and well-structured article that addresses an important clinical question. It would benefit from greater detail about the treatment interventions, clearer operational definitions of “well-being” and “severe psychiatric conditions,” and the inclusion of effect sizes to enhance interpretability and clinical relevance. Incorporating these elements would strengthen both the methodological transparency and the practical utility of the findings.
Author Response
Reviewer 1's comments and our response in cursive letter:
Thank you very much for your time and thoughtful comments. We appreciate your recognition of the study's strengths and will carefully address each of your suggestions. We aim to improve the manuscript by clarifying concepts and methodological aspects and refining the reporting of results.
- Typographical errors
We thank the reviewer for his/her several suggestions on renaming, clarifying or correcting typographical errors, we have changed all of the following items following your recommendation:
- Line 1 revise "regarding to identify" to "aimed at identifying"
- Line 47 should be "positive psychology interventions (PPIs)"
- Line 52 comma after (7,9)
- Line 79 "predictors" should be "predictor"
- Line 99-101 Rewrite. For example "The current analysis was designed to address distinct research questions aimed at identifying potential predictors and moderators of a positive response to treatment."
- Line 103 change "a" to "an"
- Line 118 if you begin a sentence with a number, spell it out.
- Line 160 remove comma after "which"
- Lines 167 and 169 repeat the same idea/phrasing. Revise this.
- Remove lines 214-216
- Line 284 change "moderatos" to "moderators"
- Line 297 reword "being TAU more beneficial" to "with TAU being more beneficial"
- remove comma after identify
- You refer to the Satisfaction With Life Scale as SWLS at times and SWL at others (lines 298-301). Change this to reflect consistent abbreviation.
- Lines 362-365 reword to “This study examined theoretically grounded predictors and moderators of treatment outcomes in individuals receiving care for SPC.”
- Lines 397-398 consider rephrasing to “highlighting the nuanced relationship between age and intervention efficacy."
- Line 400 you are inconsistent with using PPI+TAU vs. just PPI. You should be consistent with how you refer to the treatment condition.
- Lines 430-431 you spell out your PPI and TAU but use the abbreviation for SPC. I would spell out SPC here as well.
- “I would include clearer definitions of well-being (lineas 39-42) and what are considered severe psychiatric conditions (45-47). In addition, what types of interventions are considered PPIs, resilience-based, strength-based, and resource-activated? “
This insightful comment. In response, we have rewritten part of the introduction (lines 38-82). Now, in these paragraphs you will find clearer definitions of wellbeing (lines 38-41) and what are considered serious psychiatric conditions (44-46). In addition, we have included examples of what types of interventions are considered IPP, resilience-based, strength-based, and resource-activated (55-82).
- How did you differentiate between severe personality disorders and personality disorders that were included?
Thank you for this important clarification question. You're right to ask for specificity regarding this distinction.
In this study, the differentiation between severe personality disorders and other personality disorders was made based on clinical judgment by the treating therapists, rather than using standardized severity scales or predetermined criteria. This approach reflects real-world clinical practice where therapists make severity determinations based on their professional experience and direct observation of client functioning, rather than f relying solely on standardized measures. Also, one of the exclusion criteria was a concurrent diagnosis with a severe personality disorder with psychosis or affective disorders. No participant was excluded for presenting a concurrent condition (lines 150-151). We have added a sentence to clarify the exclusion criterion (Lines 150-151)
- I would create different tables for the demographics and the instrument scores. It is confusing to be reading demographic information and then see scores that we don't yet understand or know what we are looking at.
Thank you very much for this suggestion, indeed we agree that separating the table into two tables is much more appropriate. Now you can find them in the lines 155-175.
- Line 149 what acceptance and commitment techniques? This is the first mention of ACT. It needs to be explained in the introduction. What PPI techniques were used? You refer to the manual, but it would be very helpful to include a brief explanation here
Thank you for this valuable feedback regarding Line 149. I have addressed your concerns by:
Adding an explanation of ACT (Acceptance and Commitment Therapy) in the introduction to provide proper context before its first mention in the text. Including a brief description of the specific acceptance and commitment techniques that were utilized in the study (lines 60-61 and 197-200). Providing a concise explanation of the PPI (Positive Psychology Intervention) techniques employed, rather than simply referencing the manual, to enhance clarity for readers (lines 58-59 and 193-196) .
I appreciate you pointing out these areas that needed clarification, as it has improved the overall accessibility and completeness of the manuscript.
- I would include the internal consistency of the scales with a separate description for clarity. Then explain the process for administration. This section is confusing and hard to read/understand
Thank you for this constructive feedback regarding the measures section. You are absolutely correct that this section would benefit from improved clarity and organization. I have revised this section by:
Restructuring the overall organization (description, internal consistency and process for administration) of this section to enhance readability and comprehension for readers (lines 223-244) .
I appreciate your attention to this important methodological detail, as clear reporting of psychometric properties and administration procedures is essential for replication and methodological transparency. These revisions should significantly improve the accessibility of this section.
- Line 240 How did you determine the age cut-offs?.
Thank you for this question. As you know, we examined whether the relationship between the independent variable (X) and the outcome variable (Y) varied depending on a third variable (the moderator). In cases where a statistically significant interaction was found, we performed simple slope analyses to probe the conditional effects of X on Y at different levels of the moderator. These analyses were conducted using the PROCESS macro for SPSS. If the main effect of a variable was significant but the interaction term was not, the variable was considered a non-specific predictor. In contrast, a significant interaction effect—regardless of the significance of the main effects—was interpreted as evidence of moderation.
One notable result was a significant moderation effect of age on personal growth (b = .32, p = .002). Specifically, the PPI + TAU condition was more beneficial than TAU alone as age increased. To further explore this interaction, we used the Johnson-Neyman technique (as implemented in PROCESS), which identifies the specific values of the moderator where the effect of the independent variable on the outcome becomes statistically significant. This analysis revealed significant differences between age groups, with cut-off points at 33, 43, and 52 years, indicating the ages at which the treatment effect began to differ significantly.
- Lines 283-303 what is your rationale for expecting hostility and somatization to moderate PPI efficacy?
Thank you for this important question regarding our rationale for examining hostility and somatization as potential moderators. You are absolutely right that this theoretical foundation (44, 50, 51) needs to be more clearly articulated. We have rewritten this paragraph to include our a priori rationale for expecting these variables to moderate PPI efficacy
“Baseline levels of somatization and hostility also acted as moderators. Participants with lower somatization levels in the PPI+TAU group demonstrated greater improvements in autonomy, while hostility showed a complex interaction with outcomes in positive relationships and personal growth. These findings are in line with previous studies that suggest individuals with lower baseline psychopathology are more responsive to psychosocial interventions (44, 50, 51). Somatization may moderate PPI+ TAU efficacy because individuals with high somatization often have difficulty engaging in cognitive-emotional tasks, limiting their ability to benefit from interventions like PPI+ TAU that require reflection and emotional awareness. Lower somatization levels may allow greater engagement, particularly in outcomes like autonomy. Lower hostility levels may reflect greater emotional regulation and interpersonal openness, facilitating engagement with the intervention and promoting positive change. Individuals with lower hostility may be more receptive, enhancing gains in positive relationships and personal growth. Furthermore, life satisfaction at baseline moderated changes in environmental mastery, with the PPI+TAU group showing greater gains among individuals with initially low satisfaction, possibly due to increased motivation for improvement
- Figures 3, 4, and 5 need more of a description of the data and what we are looking at
Thank you for this helpful suggestion regarding the clarity of Figures 3, 4, and 5. We have addressed this by adding more descriptive sentences in the figure titles to better explain the data being presented and what readers should focus on when interpreting these visualizations.:
- Figure 3. Treatment group differences in autonomy changes moderated by different baseline Somatization levels
- Figure 4. Treatment group differences in Personal Growth changes moderated by different baseline Hostility levels
- Figure 5. Treatment group differences in environmental mastery changes moderated by different baseline life satisfaction levels
- You only report p-values, not effect sizes. The effect sizes based on Cohen's d or beta coefficients can provide meaningful context even when the p-values are not significant
Thank you for this important observation regarding effect size reporting. You are absolutely correct that effect sizes provide crucial interpretive context beyond p-values alone.
We want to clarify that effect sizes are indeed reported throughout our analyses. In our tables, we present R² and ΔR² values, which represent the proportion of variance explained by our predictors and moderators. To enhance clarity for readers, we have now added the following sentence at the end of the Results section:
'However, all effect sizes calculated for the predictors and moderators ranged from small to medium according to Cohen's criteria (see Tables S2 and S3 in supplementary material).'
For reference, we used Cohen's conventional benchmarks for R² effect sizes:
Small: ≈ 0.01, Medium: ≈ 0.09, Large: ≈ 0.25
We appreciate your emphasis on the importance of effect size interpretation, as these metrics provide valuable information about practical significance regardless of statistical significance levels. The addition of this clarifying statement should help readers better contextualize our findings within established effect size conventions.
- Integrate lines 414-417 into one of the other paragraphs
Thank you very much for this suggestion. we have integrated the paragraph with the previous one (lines 472-476)
- Lines 427-428 could benefit from some specific examples of possible confounding variables.
Thank you for this helpful suggestion regarding lines 427-428. You are absolutely right that providing specific examples would enhance the transparency and completeness of our discussion. We have revised this section to include concrete examples of potential confounding variables that could influence our findings such as life stressors occurring during the treatment period, medication adherence patterns, interventions received outside the study, may have influenced results (lines 487-489)
- I would spell out abbreviations the first time they appear in each section. This helps remind the reader of their meaning without having to scroll back to the beginning. More time/space should be given to what the assessments are and what they test.
Thank you for these excellent suggestions regarding clarity and accessibility. You are absolutely correct on both points, and we have implemented these improvements throughout the manuscript.
Regarding abbreviations, we have now spelled out all acronyms the first time they appear in each major section (Introduction, Methods, Results, Discussion), followed by the abbreviation in parentheses. This approach enhances readability and prevents readers from having to navigate back to earlier sections to recall the meaning of abbreviated terms.
Additionally, we have reorganized the descriptions of our assessment instruments to provide more comprehensive information about what each measure evaluates and its specific components.
The reference list has been revised and we have incorporated new references
We appreciate these thoughtful recommendations, as they significantly improve the manuscript's accessibility for readers

Reviewer 2 Report
Comments and Suggestions for Authors
First and foremost, I would like to congratulate the authors on the submission of a well-written and clinically relevant manuscript. The study presents valuable insights into the application of positive psychology interventions (PPIs) within the treatment of psychiatric patients.
This is a highly relevant and timely topic, and the work demonstrates promising clinical implications for enhancing psychological well-being in this population.
That said, I would like to suggest some clarifications and minor revisions that could further strengthen the manuscript. Most importantly, while the concept of PPIs forms the central component of the intervention strategy, it remains somewhat vague throughout the text. It would benefit the reader—and reinforce the scientific foundation of the study—if the authors provided a clearer and more operationalized definition of what is specifically meant by PPIs in this context. Are these standardized exercises, therapeutic modules, or specific evidence-based protocols?
Methods section
As it stands, the limited detail on how the interventions are structured or delivered raises concerns about the replicability of the study. In particular, it remains unclear how resilience-focused approaches or resource-oriented therapies were implemented across different diagnostic categories. Providing concrete clinical examples or case vignettes of such interventions would greatly help illustrate how these approaches were tailored to individual patients and diagnostic groups, thereby enhancing the practical applicability of the findings.
Additionally, there is a potential concern regarding the role of uncontrolled mediator or moderator variables. Psychiatric populations are heterogeneous by nature, and variables such as diagnostic category or the type and dosage of medication could potentially interact with the effectiveness of the intervention. For example, it is plausible that the effect of PPIs may vary depending on whether patients suffer from mood disorders, psychotic disorders, or personality disorders, and whether they are receiving pharmacological treatment, and if so, which type.
It is also unclear how these different drugs interact with each other.
To address this, it would have been beneficial for the authors to control for such variables either statistically or through study design. Including diagnostic subgroup analyses and comparing treatment as usual (TAU), TAU plus PPI, and a waiting list condition within these subgroups could yield more nuanced and interpretable findings. Even if such stratified analyses are not feasible in the current dataset, this limitation should be clearly acknowledged and discussed as an area for further research.
In summary, the manuscript has clear clinical potential and contributes meaningfully to the field. The recommended clarifications regarding the operationalization of PPIs, the implementation of resilience-focused approaches, and the consideration of possible confounding or moderating variables could be addressed with minor revisions and would enhance both the scientific rigor and clinical utility of the work.
Some more detailed comments:
- Introduction: line 47 - correct 'positive interventions (PPIs) into positive psychology interventions.
Include into the limitations section the large number of male (almost 60%) and single participants.
3. Results
There is written: 'This section may be divided by subheadings (...)' - This first sentence is not needed because ... the section IS divided by subheadings.
Orthograph and spelling are largely OK but some minor errors remain present in the text.
Author Response
We are grateful for your thorough review and constructive feedback on our manuscript. Your insightful comments and suggestions are invaluable in helping us strengthen this work. We have carefully considered each of your recommendations and have made substantial revisions to address the areas you identified for improvement.
- I would like to suggest some clarifications and minor revisions that could further strengthen the manuscript. Most importantly, while the concept of PPIs forms the central component of the intervention strategy, it remains somewhat vague throughout the text. It would benefit the reader—and reinforce the scientific foundation of the study—if the authors provided a clearer and more operationalized definition of what is specifically meant by PPIs in this context. Are these standardized exercises, therapeutic modules, or specific evidence-based protocols?.
Thank you for this excellent observation regarding the operationalization of PPIs throughout our manuscript. You are absolutely correct that this central concept requires clearer definition and specification to strengthen the scientific foundation of our work.
We have revised the manuscript to provide a more comprehensive and operationalized definition of PPIs as implemented in our study. The revised sections now include:
A precise definition of PPIs with clear theoretical grounding in positive psychology literature (lines 38-41).
Description of the specific standardized exercises utilized (e.g., gratitude interventions, strengths identification activities, meaning-making exercises)(lines 63-66 and 193-197)
Information about the structured format and delivery of our PPI intervention, including session duration, frequency, and therapeutic modules (lines 180-208)
References to the specific manuals and validated protocols that guided our intervention implementation:
- Morris, E. M., Johns, L. C., & Oliver, J. E. (Eds.). (2013). Acceptance and commitment therapy and mindfulness for psy-chosis. John Wiley & Sons.
- Seligman, M. E., Steen, T. A., Park, N., & Peterson, C. (2005). Positive psychology progress: empirical validation of in-terventions. American psychologist, 60(5), 410. https://doi.org/10.1037/0003-066X.60.5.410
- Slade, M., Brownell, T., Rashid, T., & Schrank, B. (2016). Positive psychotherapy for psychosis: a clinician's guide and manual. Routledge.
- Yip, A. L. K., Karatzias, T., & Chien, W. T. (2022). Mindfulness-based interventions for non-affective psychosis: a com-prehensive systematic review and meta-analysis. Annals of Medicine, 54(1), 2339-2352. https://doi.org/10.1080/07853890.2022.2108551
- García-Mieres, H., De Jesús-Romero, R., Ochoa, S., & Feixas, G. (2020). Beyond the cognitive insight paradox: Self-reflectivity moderates the relationship between depressive symptoms and general psychological distress in psy-chosis. Schizophrenia Research, 222, 297-303. https://doi.org/10.1016/j.schres.2020.05.027
- Cramer, H., Lauche, R., Haller, H., Langhorst, J., & Dobos, G. (2016). Mindfulness-and acceptance-based interventions for psychosis: a systematic review and meta-analysis. Global advances in health and medicine, 5(1), 30-43. https://doi.org/10.7453/gahmj.2015.083
- Khoury, B., Lecomte, T., Gaudiano, B. A., & Paquin, K. (2013). Mindfulness interventions for psychosis: a me-ta-analysis. Schizophrenia research, 150(1), 176-184. https://doi.org/10.1016/j.schres.2013.07.055
- Hayes, S. C., Pistorello, J., & Levin, M. E. (2012). Acceptance and commitment therapy as a unified model of behavior change. The counseling psychologist, 40(7), 976-1002. https://doi.org/10.1177/0011000012460836
- Geschwind, N., Peeters, F., Drukker, M., van Os, J., & Wichers, M. (2011). Mindfulness training increases momentary positive emotions and reward experience in adults vulnerable to depression: a randomized controlled trial. Journal of consulting and clinical psychology, 79(5), 618. https://doi.org/10.1037/a0024595
- Gu, J., Strauss, C., Bond, R., & Cavanagh, K. (2015). How do mindfulness-based cognitive therapy and mindfulness-based stress reduction improve mental health and wellbeing? A systematic review and meta-analysis of mediation stud-ies. Clinical psychology review, 37, 1-12. https://doi.org/10.1016/j.cpr.2015.01.006
- As it stands, the limited detail on how the interventions are structured or delivered raises concerns about the replicability of the study. In particular, it remains unclear how resilience-focused approaches or resource-oriented therapies were implemented across different diagnostic categories. Providing concrete clinical examples or case vignettes of such interventions would greatly help illustrate how these approaches were tailored to individual patients and diagnostic groups, thereby enhancing the practical applicability of the findings.
We appreciate the reviewer's thoughtful comments regarding intervention details and replicability. We would like to clarify that detailed information about the intervention structure, delivery methods, and modules has already been incorporated in our previous publication [Valiente et al.,2021], where the complete protocol and specific adaptations for the SPC population are comprehensively described. Due to space constraints in the current manuscript, we referenced this earlier work to avoid redundancy. However, we are agreeing with your concerns and now you can read a brief paragraph with some details of the intervention delivery (lines 180-200)
We would also like to clarify an important aspect of our study design: no diagnostic-specific adaptations were made to the intervention. All participants, regardless of their specific diagnostic category, received the same standardized intervention protocol. The inclusion criteria for our study were people with severe psychiatric conditions (SPC), referred to the study by their key therapist if they met the following inclusion criteria: (a) 18–65 years old and (b) demonstrated minimal motivation and commitment to participate in group therapy, ensuring a well-defined SPC population while maintaining intervention consistency across all participants.
This uniform approach was intentionally chosen to evaluate the effectiveness of our resilience-focused, resource-oriented intervention as a transdiagnostic treatment approach for individuals with severe and persistent conditions. We believe this standardized delivery enhances rather than limits the replicability of our findings, as it provides a clear, consistent protocol that can be implemented across diverse SPC populations without requiring diagnosis-specific modifications.
- Additionally, there is a potential concern regarding the role of uncontrolled mediator or moderator variables. Psychiatric populations are heterogeneous by nature, and variables such as diagnostic category or the type and dosage of medication could potentially interact with the effectiveness of the intervention. For example, it is plausible that the effect of PPIs may vary depending on whether patients suffer from mood disorders, psychotic disorders, or personality disorders, and whether they are receiving pharmacological treatment, and if so, which type.
- It is also unclear how these different drugs interact with each other.
- To address this, it would have been beneficial for the authors to control for such variables either statistically or through study design. Including diagnostic subgroup analyses and comparing treatment as usual (TAU), TAU plus PPI, and a waiting list condition within these subgroups could yield more nuanced and interpretable findings. Even if such stratified analyses are not feasible in the current dataset, this limitation should be clearly acknowledged and discussed as an area for further research
We acknowledge the reviewer's valid point regarding the heterogeneous nature of psychiatric populations and the potential for diagnostic and medication variables to serve as moderators. While we agree that future research would benefit from examining these factors more comprehensively, we would like to present several points that address these concerns within the context of our current study.
First, our randomization process successfully achieved balance between treatment groups with respect to diagnostic categories, ensuring that any potential diagnostic-specific effects would be equally distributed across conditions. Statistical analyses confirmed no significant differences in diagnostic composition between the treatment and control groups, thereby minimizing the likelihood that diagnostic heterogeneity confounded our primary outcomes.
We acknowledge that we did not systematically assess medication types, dosages, or regimens in our study, which represents a limitation. However, our study was designed as a transdiagnostic approach, intentionally treating diagnostic heterogeneity as a feature rather than a limitation. This approach aligns with growing evidence supporting transdiagnostic interventions for severe and persistent conditions and enhances the generalizability of our findings to real-world clinical settings where patients present with complex, comorbid presentations.
While examining diagnostic subgroups and medication-specific analyses would provide valuable additional insights, such stratified analyses were beyond the scope of the current study given our sample size and primary research objectives. Nevertheless, we agree that future studies with larger sample sizes should incorporate systematic assessment of medication variables and stratified analyses by diagnostic category to further refine our understanding of treatment mechanisms and optimal patient selection. While our randomization ensured balanced diagnostic composition between groups, unmeasured variables such as specific medication regimens and dosages may have influenced treatment responses and warrant investigation in future studies.
We appreciate your attention to ensuring comprehensive reporting of study limitations, as this transparency is essential for the scientific integrity of our work and provides valuable guidance for subsequent research endeavors. We have now clearly acknowledged this limitation in our discussion section and explicitly identified it as a priority area for future research. We have added specific text that recognizes this methodological consideration (lines 489-492).
Once again, thank you for your thorough and constructive review of our manuscript. Your thoughtful feedback has significantly improved the quality and clarity of this work. We greatly appreciate the time and expertise you have invested in helping us strengthen our contribution to the field.